# Assessing Impacts of Land Subsidence in Victoria County, Texas, Using Geospatial Analysis

Muhammad Younas [1], Shuhab D. Khan [1], Muhammad Qasim [1,2] and Younes Hamed [3,*]

1   Department of Earth and Atmospheric Sciences, Science and Research Building 1, University of Houston, 3507 Cullen Blvd, Room 312, Houston, TX 77204, USA
2   Geoscience Advanced Research Laboratories, Geological Survey of Pakistan, Islamabad 44230, Pakistan
3   Laboratory for the Application of Materials to the Environment, Water and Energy (LAM3E), Department of Earth Sciences, University of Gafsa, Gafsa 2112, Tunisia
*   Correspondence: hamedhydro.tn@gmail.com; Tel.: +216-98-910-693

**Abstract:** Land subsidence is an ongoing problem negatively affecting Victoria County along the Gulf Coast. Groundwater withdrawal and hydrocarbon extraction in the County are some of the known factors behind this geological hazard. In this study, we have used geospatial analysis and a conceptual model to evaluate land subsidence. A significant decline in the groundwater level in this area was noted from 2006 to 2016. The decline in the water level correlates with the major drought events along the Gulf Coast reported in earlier studies. These results are further corroborated by the emerging hotspot analysis performed on the groundwater data. This analysis divides the study area into intensifying, sporadic, and persistent hotspots in the northwest region and intensifying, persistent coldspots in the southeast region of Victoria County. Hydrocarbon production data show high oil and gas extraction from 2017 to 2021. There are a higher number of hydrocarbon production wells in the central and southern regions of the County than elsewhere. The conceptual models relate these events and suggest the existence of subsidence in the County, through which the water and hydrocarbon reservoirs in the study area may lose their reservoir characteristics due to sediment compaction.

**Keywords:** land subsidence; groundwater; optimized hotspot; emerging hotspot; conceptual model





## 1. Introduction

The sinking of the ground surface in comparison with the surrounding land or the sea level, termed land subsidence, is one of the geohazards affecting both continental and coastal areas of many countries around the globe. Among many natural and anthropogenic processes contributing to land subsidence are tectonic motion, sea-level rise, and excessive withdrawal of natural resources such as coal, oil, or gas. The excessive withdrawal of groundwater is one of the foremost reasons for land subsidence in Houston, Texas [1], Mexico [2], Tunisia [3–5], Algeria [6], Thailand [7], Italy [8,9], and China [10]. According to some surveys, more than one-third of the world's population lives in coastal regions [11]. These coastal areas are more vulnerable than inland areas to natural and/or anthropogenic hazards such as flooding, hurricanes, changes in the sea level, climate change, dam construction, and overexploitation of mineral resources [12–16]. In addition to the Houston Metroplex, the coastal part of Texas is also suffering from land subsidence, thus exposing the coastal communities of this region to serious geological hazards [17]. Factors behind the subsidence in this area include excessive groundwater withdrawal, hydrocarbon over-extraction, growth faults, and sediment compaction [1,18–20].

Geospatial analysis is a valuable technique that allows for solving complex, location-oriented problems by integrating several spatial layers that are somehow related to the problem. It also generates maps that help visualize the spatial variations with the application of different spatial operators on the input layers, which ultimately assist in improving problem management and decision-making. Spatial analysis has been used in

many ways in several fields, such as soil sciences [21], groundwater studies [22,23], medical sciences [24–26], transportation [27,28], e-commerce, and business [29,30]. The emerging hotspot analysis is an innovative spatial analysis tool that can detect spatial-temporal trends and varying patterns of different phases in each period; it has been progressively utilized in multiple scientific disciplines [31–33]. This geospatial technique uses a combination of two statistical measures, the Getis-Ord Gi statistic to identify the location and degree of spatial clustering and the Mann–Kendall trend test to evaluate temporal trends across the time series [34]. The conceptual model is an illustrative image of the groundwater system in terms of hydrogeologic elements, system boundaries that include inputs and outputs varying with time, and hydrodynamic and transport properties, as well as spatial variability [35,36].

Land subsidence is an ongoing problem negatively affecting Victoria County along the Gulf Coast. A recent publication on subsidence in the Texas Coastal Bend reported an average subsidence rate of −7.55 to 5.51 mm/yr [37]. The study used interferometric synthetic aperture radar (InSAR) data from the Sentinel-1 satellite from October 2016 to July 2019 and revealed the highest subsidence of −15 mm/yr in the south of Victoria [37]. Groundwater withdrawal and hydrocarbon extraction in the County are possible causes of this subsidence [37].

This research work integrated multiple datasets, i.e., groundwater, hydrocarbon production, and conceptual models, to investigate and quantify the role of these factors in land subsidence in Victoria County. Knowing the factors involved in subsidence, and their role in the damage caused to buildings and other infrastructure, will help mitigate future damage and help us plan possible ways to overcome and minimize risk in this region.

## 2. Materials and Methods

### 2.1. Materials

Victoria County is situated on the coastal plains of Texas, around 50 miles from the Gulf of Mexico and 20 miles from the adjacent bay waters. According to the Web Soil Survey of the United States Department of Agriculture, the topography of this area is flat, with an average elevation of 95 feet above mean sea level. The coastal region is located on a passive depositional margin, with the geologic formations primarily composed of alternating sand, clay layers, and silt, ranging in age from the Miocene to the Holocene [38]. Quaternary deposits and the Neogene sedimentary rocks dominate the study area. Victoria County is drained by a hydrological network in a northwest–southeast direction (Figure 1).

The hydrogeological setup of the study area is dominated by the Gulf Coast aquifer system, comprised of three main units: the Jasper, the Evangeline, and the Chicot aquifers corresponding in age to the Miocene, Pliocene, and Pleistocene/Holocene, respectively (Figure 2). These aquifers are thick, unconfined, or semi-confined and dip toward the Gulf of Mexico [39]. The Guadalupe River passes through Victoria County, and Green Lake is situated in the southern part of this County. The changes in early sea levels, depositional volumes, and sources led to disjointed sands, gravel, silt, and clay beds. The subsidence of the depositional basin and growing land surfaces produced many of the Gulf Coast aquifer units that increase in the thickness downdip. This variation, along with growth faults in this area, factored into the heterogeneity currently seen in various strata of the Gulf Coast aquifer system [40].

The multiple datasets used in this study included groundwater well data, hydrocarbon production, and conceptual models. Groundwater data for 95 wells were obtained from the Texas Water Development Board (TWBD) [41]. Oil and gas production data were obtained from the Railroad Commission of Texas, while the oil and gas data for 1932 wells were obtained from the Homeland Infrastructure Foundation Level [42].

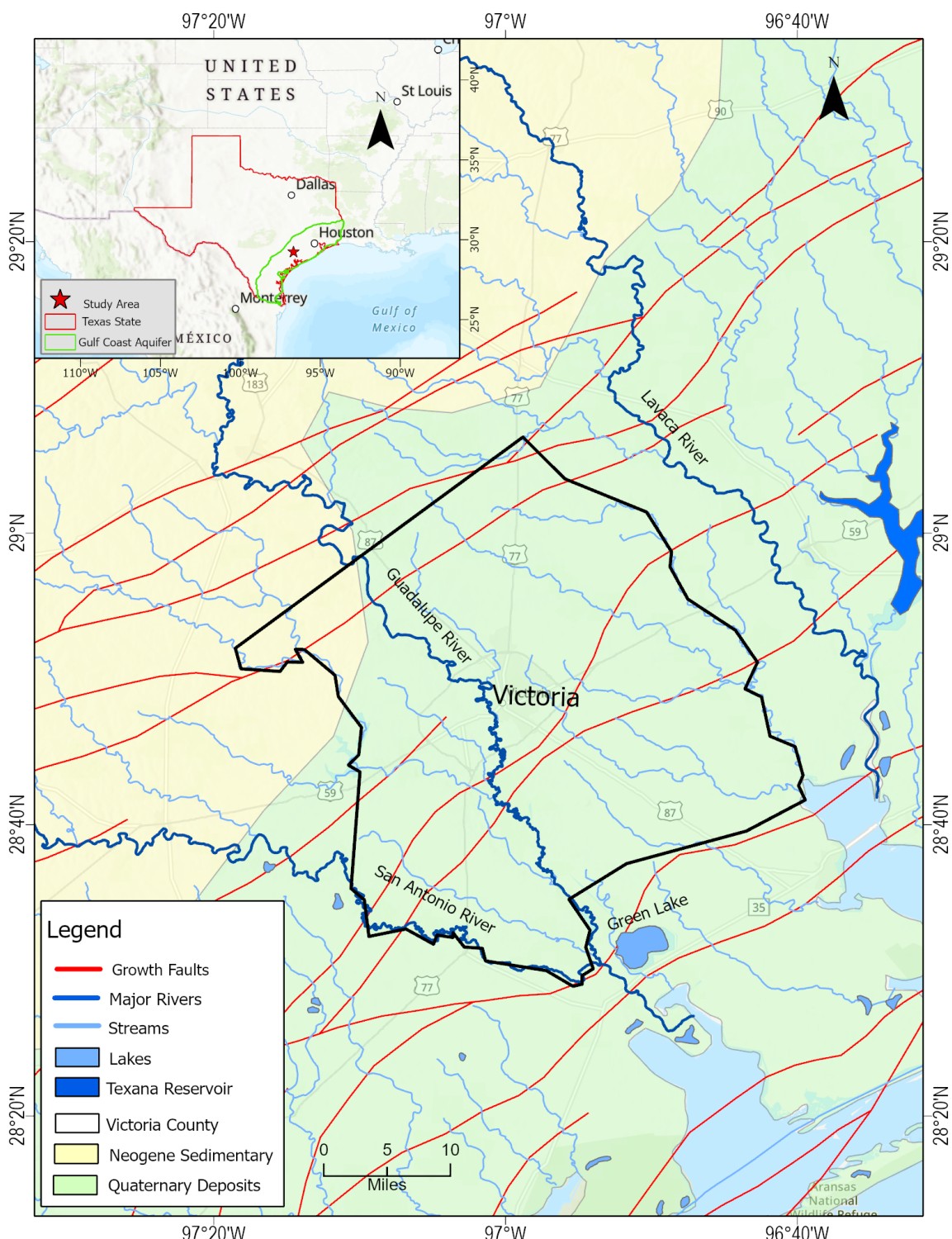

**Figure 1.** Map showing the surface geology, hydrological network, and the trend of growth faults in and around the study area.

| System | Series | Stratigraphic Units | | | Hydrogeology Potential |
|---|---|---|---|---|---|
| Quaternary | Holocene | Alluvium | | | Chicot Aquifer |
| | Pleistocene | Beaumont Clay | | | |
| | | Lissi Formation | Montgomery Formation | | |
| | | | Bentley Formation | | |
| | | Willis Sand | | | |
| | Pliocene | Goliad Sand | | | Evangeline Aquifer |
| Tertiary | Miocene | Fleming Formation/Lagarto Clay | | | Burkeville Confining System |
| | | Oakville Sandstone | | | Jasper Aquifer |
| | Oligocene | Catahoula Tuff or Sandstone | Upper part of Catahoula tuff | | Catahoula Confining System |
| | | | Anahuac Formation | | |
| | | | Frio Formation | | |
| | | Frio Clay | Vicksburg Group Equivalent | | |

**Figure 2.** Stratigraphic succession and the hydrostratigraphic divisions for corresponding stratigraphic units (modified from [40]).

*2.2. Methods*

The methodology adopted for this study is shown in Figure 3. This study used advanced geospatial analysis tools on groundwater well data, locations of oil and gas wells, and production data. This work generated optimized and emerging hotspots to find statically significant clusters, detailed below.

The groundwater data for Victoria County were used to produce potentiometric maps. These maps are a valuable tool used by hydrogeologists to deduce useful information such as the groundwater flow direction and hydraulic conductivity variations [43–45]. After initial data processing, the potentiometric surface maps were prepared using the inverse distance weighting (IDW) interpolation method in ArcGIS Pro to illustrate the groundwater level variations in the County from 2000 to 2021. The IDW is built on the concept of Tobler's first law, which states that everything is associated with everything else, but adjacent things are more interrelated than distinct things [17,44]. The functionality of IDW is based on the formula presented in Equation (1) [45].

$$V = \frac{\sum_{i=1}^{n} v_i \frac{1}{d_i^p}}{\sum_{i=1}^{N} \frac{1}{d_i^p}} \tag{1}$$

where $d$ is the distance between prediction and measurement points, $v_i$ is the measured parameter value, and $p$ is a power parameter.

Clustering techniques in data mining, such as density maps (point density and kernel density), highlight the clusters in a given set of data without defining if the clusters are statistically significant [46,47]. Clustering of hydrocarbon data was performed using hotspot analysis, which identifies the locations of statistically significant hotspots (the areas of high activity) and coldspots (the areas of lower activity) in the data by producing

z-score and *p*-values [48]. The Getis-Ord Gi* statistic [49,50] used in the process is given in Equations (2)–(4):

$$G_i^* = \frac{\sum_{j=1}^{n} w_{i,j} x_j - X^- \sum_{j=1}^{n} w_{i,j}}{S \sqrt{\frac{[\sum_{j=1}^{n} w_{i,j}^2 - (\sum_{j=1}^{n} w_{i,j})^2]}{n-1}}} \tag{2}$$

where $x_j$ represents the attribute value of feature $j$, $w_{i,j}$ is the spatial feature $i$ and $j$, $n$ is equal to the total number of features, and:

$$X^- = \frac{\sum_{j=1}^{n} x_j}{n} \tag{3}$$

$$S = \sqrt{\frac{\sum_{j=1}^{n} x_j^2}{n} - (X^-)^2} \tag{4}$$

The Gi* statistic is a z-score so no supplementary calculations are needed. This tool makes a new output feature class with a z-score, *p*-value, and confidence level bin for every feature in the input feature class. The groundwater data also use the emerging hotspot analysis, which identifies the trends in the data over a timespan. The space-time cube, a mandatory component required for running an emerging hotspot analysis, is produced using the groundwater well data. This cube is a descriptive statistic contained in bins, with the base of every bin representing a geographical location with x and y values and the height representing time with a z value [51]. A neighborhood distance of 8 km was used in creating the cube.

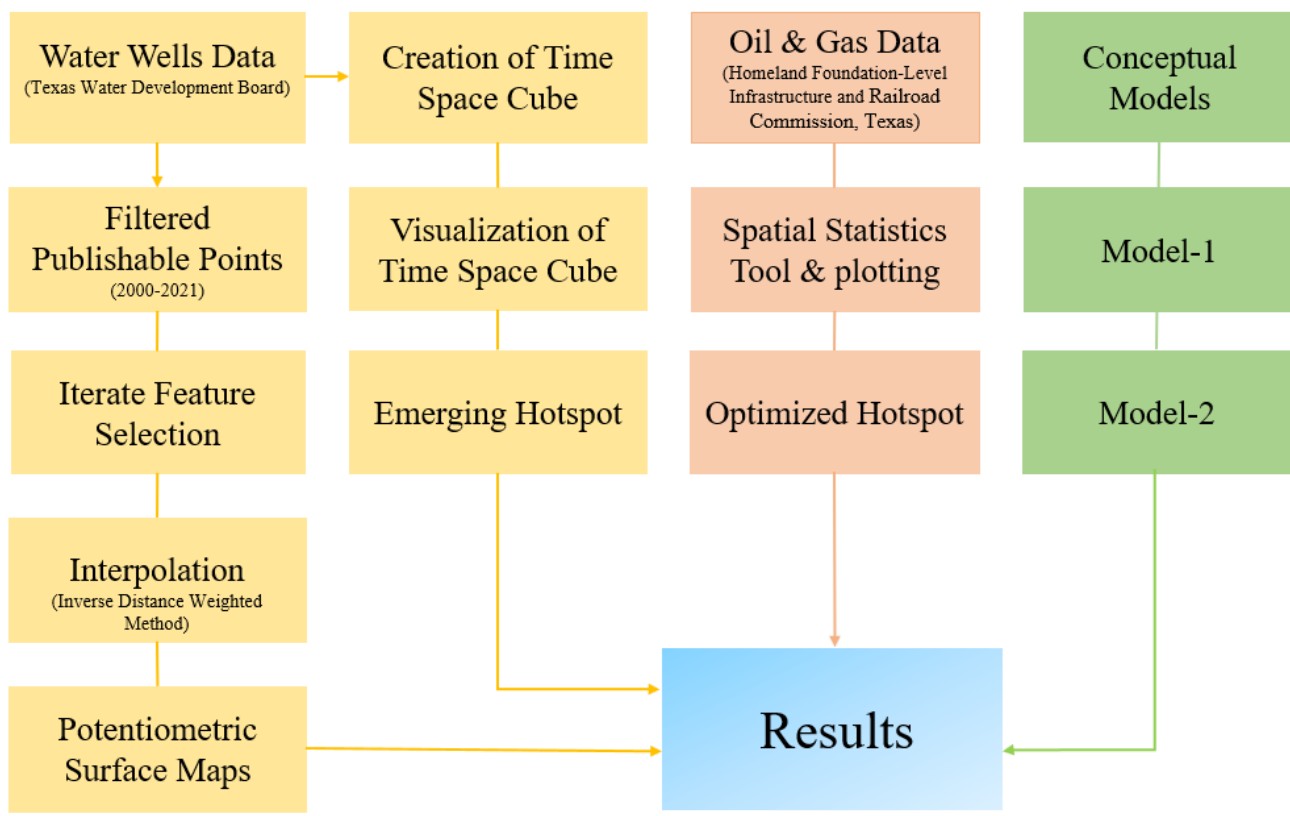

**Figure 3.** Methodological framework showing the data used and the main steps implemented in this study.

Additionally, optimized hotspot analysis was performed on the hydrocarbon well data to find the areas of high oil and gas pumping activity. This clustering technique is an extension of the Getis-Ord Gi* hotspot analysis, which finds the spatial clusters of statistically significant higher values (hotspots) or lower values (coldspots) as compared to the surrounding locations in an area using the parameters derived from characteristics of the input data [52]. Many researchers have increasingly used this statistical method in many fields [53–58].

Based on the data available and the results obtained, two groundwater conceptual models, Model-1 and Model-2, were proposed for the Gulf Coast aquifer system in Victoria County. Such conceptual models were designed based on knowledge of the study area. Development of these models for an aquifer requires the rigorous and meticulous integration of data, information, and reports relating to aquifers within a study area and groundwater movements [59,60]. The choice of the geological context and other features in the development of the conceptual model is critical in determining the consequences and may have noteworthy monetary results for areas scheduled for development [61]. The rudimentary components adopted for developing these conceptual models are physical and appropriate boundary conditions, hydrodynamic properties, and surface and subsurface water circulation passages. The conceptual models subsequently developed have characteristics such as recharge/discharge zones, groundwater levels, and connection between sediment compaction, subsidence, and withdrawal points.

## 3. Results

### 3.1. Potentiometric Surface Maps

Potentiometric maps of Victoria County showed that the annual groundwater level from 2000 to 2021 had a common trend of a high potentiometric surface in the northwest of the County, decreasing toward the southeast. This trend correlates with the flow of the rivers, including the Guadalupe River, running through the County. The deepest groundwater level of ~28–32 m was observed over a wide area in the northwest of the County from 2000 to 2003 (Figure 4a–d). However, it was not constant over time and the water level increased gradually from ~22 m to 20 m during the period of 2004–2009 (Figure 4e–j). The water level went down again from ~22 m to 26 m during the period of 2010–2015 (Figure 4k–p). Finally, the water level rose steadily from ~26 m to 20 m until 2021 (Figure 4q–v). The southeast region of this County had the shallowest groundwater level, much lower than that of the northwest region, but with a notable rise in groundwater from ~8 m to 4 m over the period from 2013 to 2021 (Figure 4n–v).

### 3.2. Emerging Hotspot Analysis

The emerging hotspot analysis showed two new, two intensifying, three sporadic, and four persistent hotspots in the northwest region of the study area (Figure 5). The two new hotspots were the statistically significant hotspots in the area, where groundwater level decline was observed only during the most recent times. The two intensifying hotspots showed that the groundwater level decline increased over time. The three sporadic hotspots represented transition in the groundwater level making these on-again and off-again hotspots. The four persistent hotspots had deep groundwater levels over time. There were two consecutive, one sporadic, and one oscillating coldspot in the central region of the County. The southwest part of the County showed five persistent coldspots and two intensifying coldspots, indicating that the groundwater level was shallow and getting shallower with time (Figure 5).

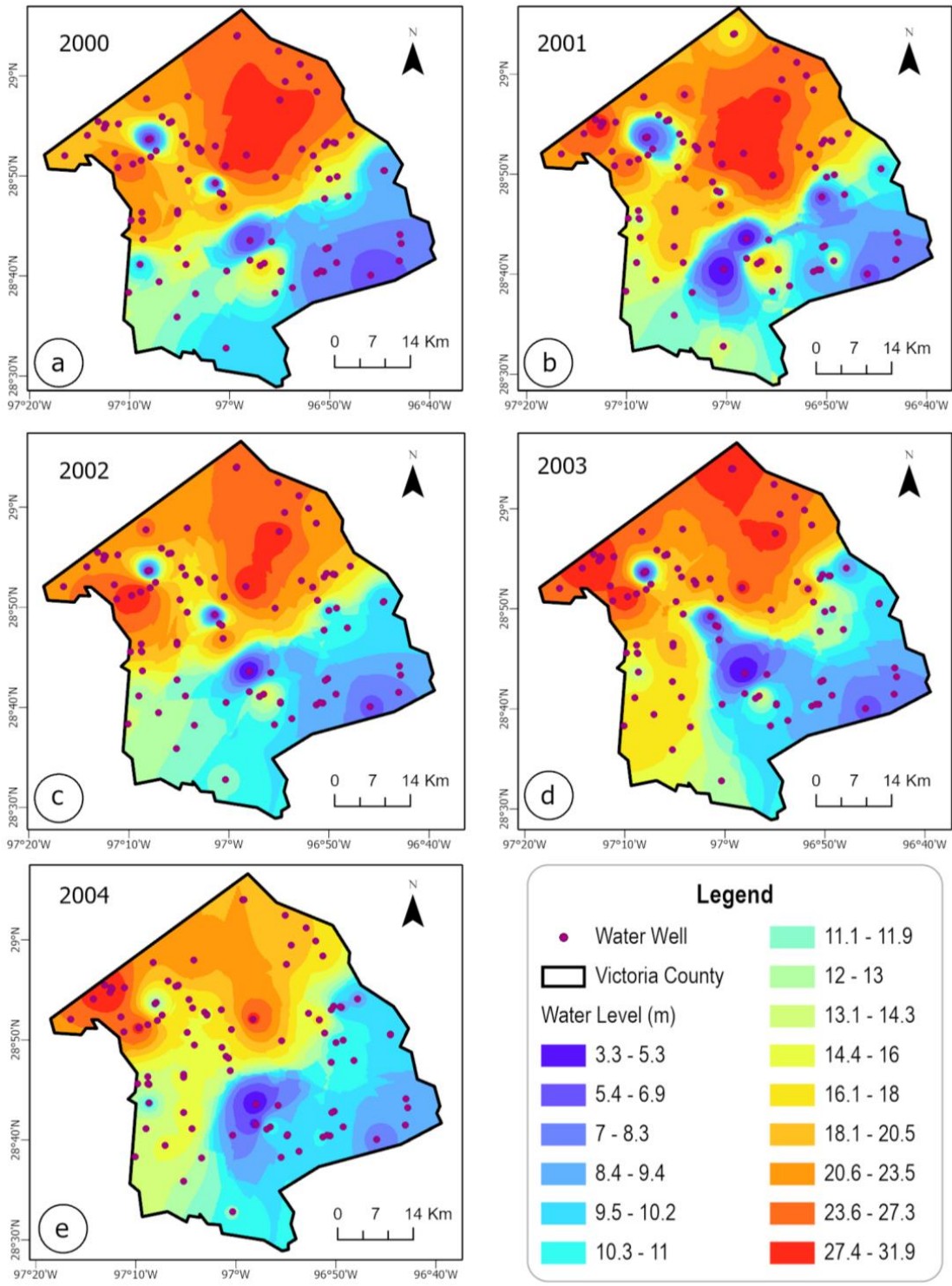

**Figure 4.** *Cont*.

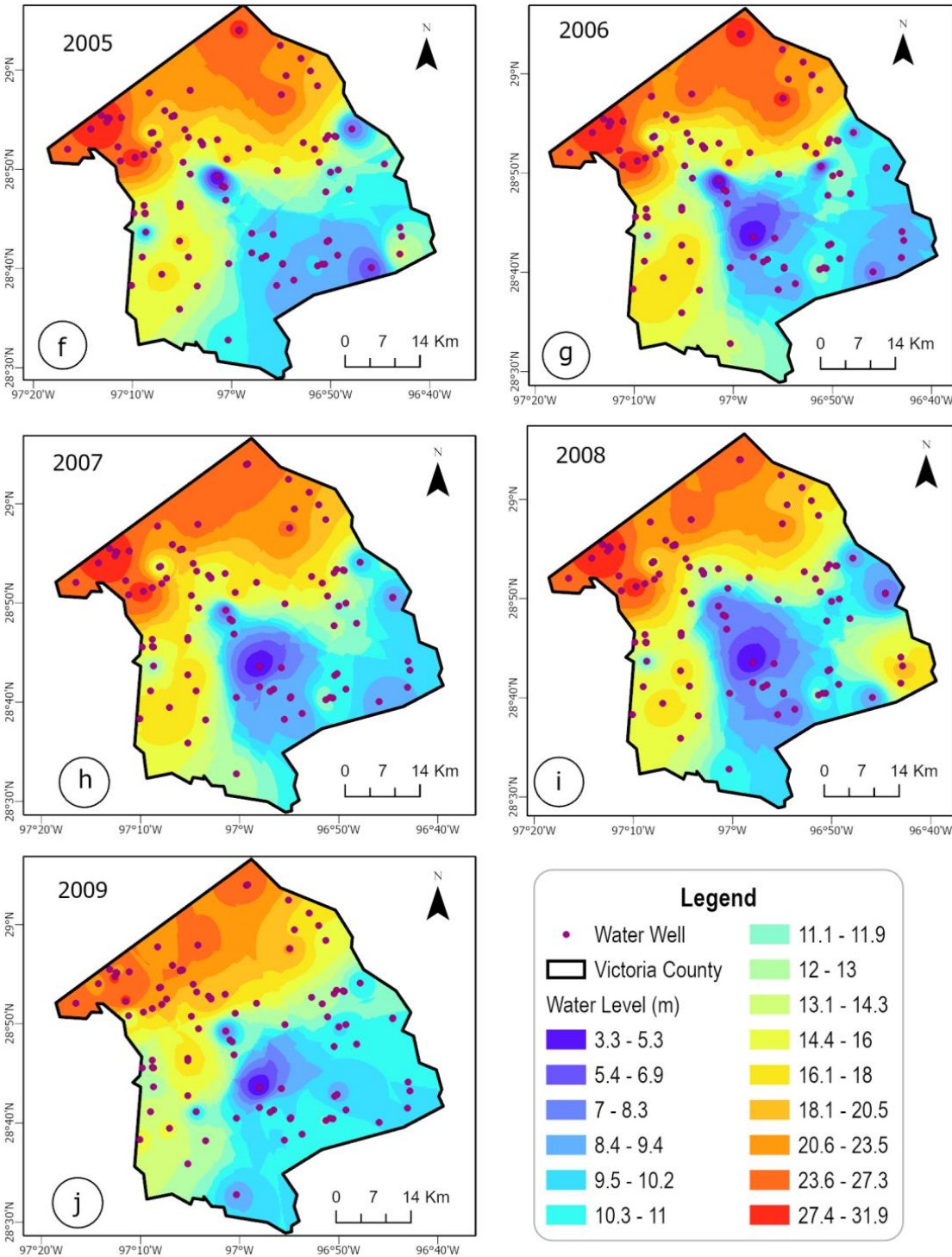

**Figure 4.** *Cont.*



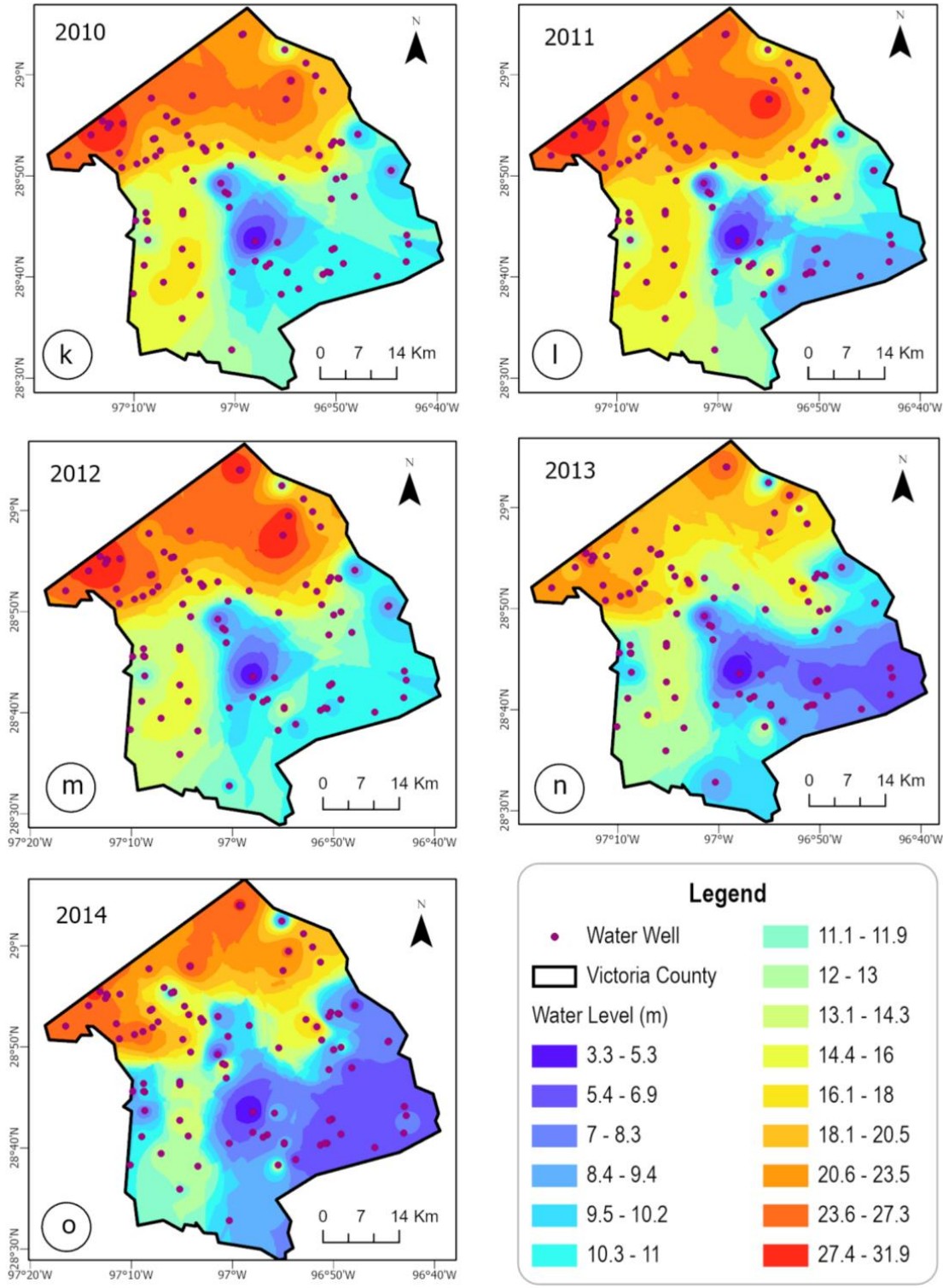

**Figure 4.** *Cont.*

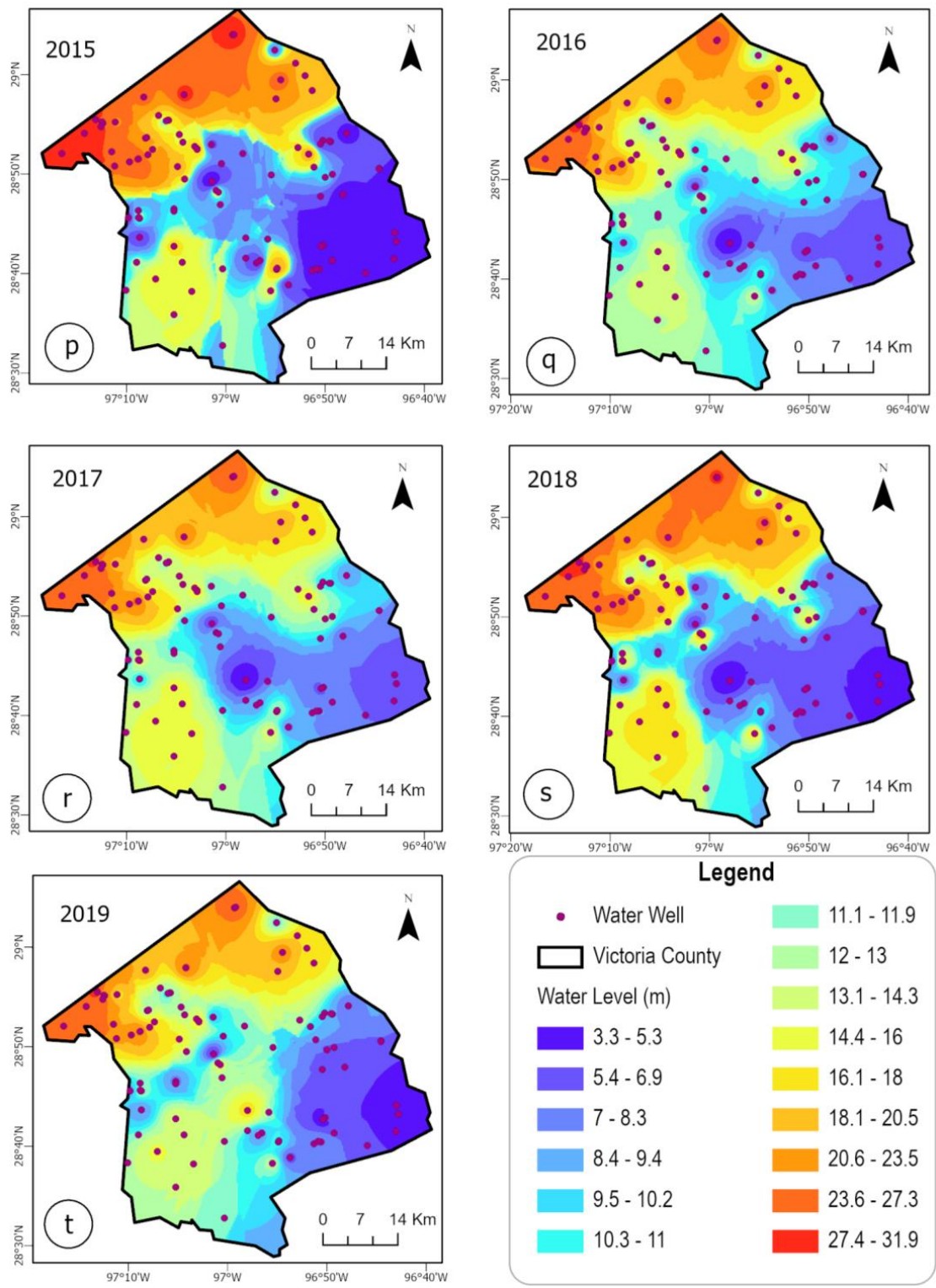

**Figure 4.** *Cont.*

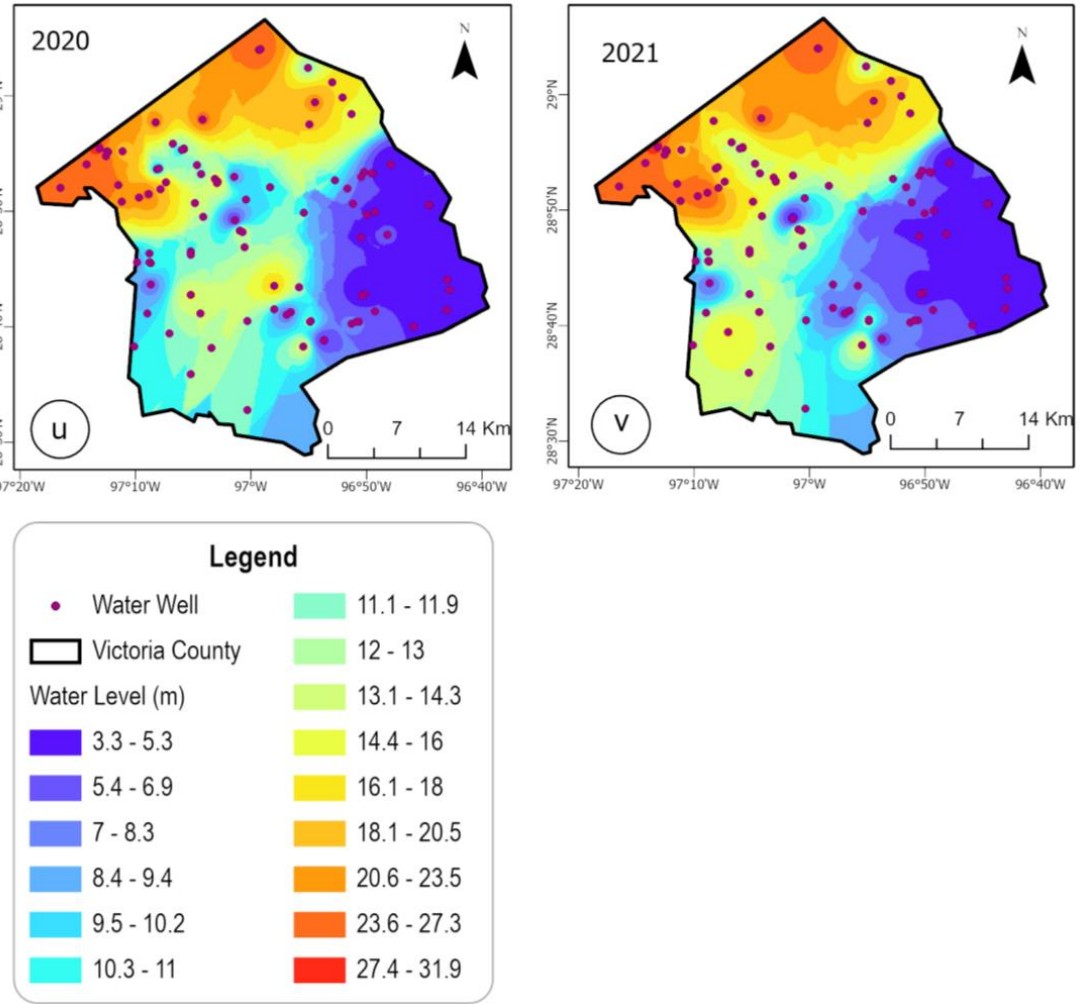

**Figure 4.** (**a–e**) Potentiometric surface maps showing the groundwater decline from 2000 to 2004. (continued). (**f–j**) Potentiometric surface maps showing the groundwater decline from 2005 to 2009. (continued). (**k–o**) Potentiometric surface maps showing the groundwater decline from 2010 to 2014. (continued). (**p–t**) Potentiometric surface maps showing the groundwater decline from 2015 to 2019. (continued). (**u–v**) Potentiometric surface maps showing the groundwater decline from 2020 to 2021.

The 3D space-time cube derived from the groundwater well data using ArcGIS Pro was translated into a graph showing the average water level (Figure 6) for Victoria County from 2000 to 2021. The deepest groundwater level of~17m was observed in 2000. However, the groundwater level became drastically shallower up to 2006. Afterward, it gradually dropped until 2010. A groundwater level drop was observed again during 2016, but this drop was not that intense. The overall pattern of the groundwater level in the County was an increase over the past 21 years with a prolonged decline effect from 2006 to 2015.

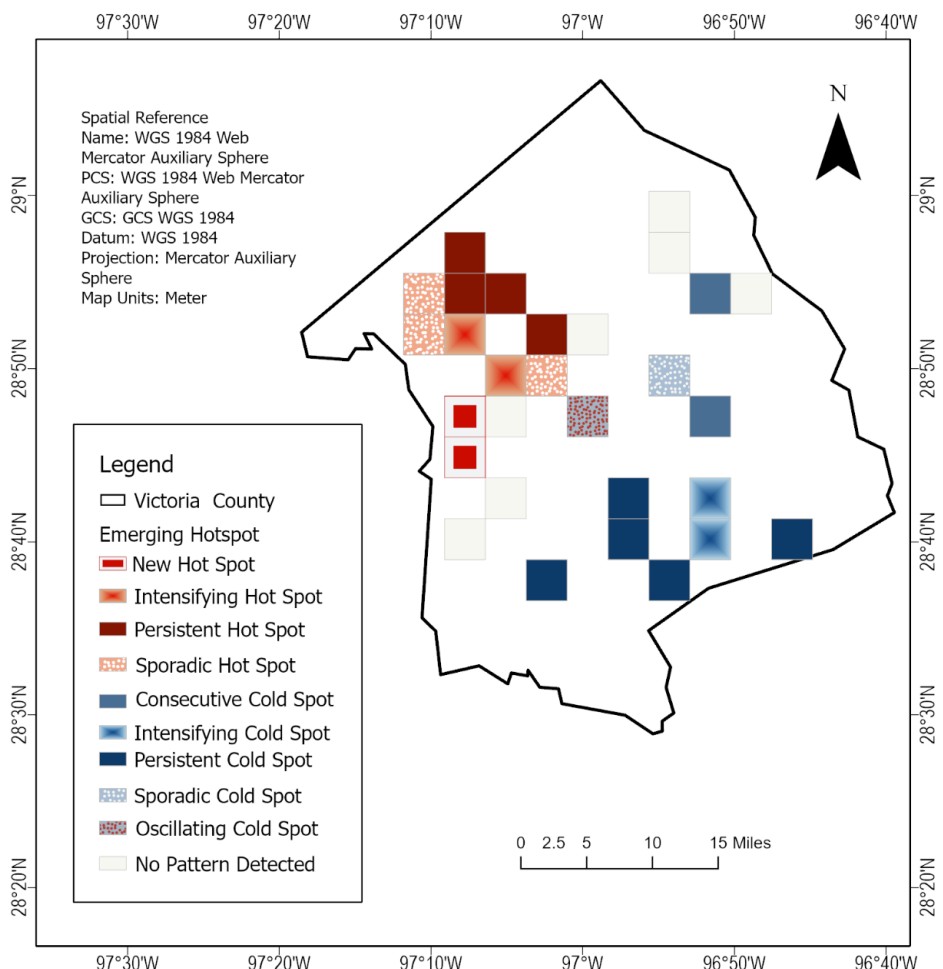

**Figure 5.** Results of emerging hotspot analysis for the groundwater transition over the study area from 2000 to 2021. Blank areas show the water well data were insufficient for the analysis.

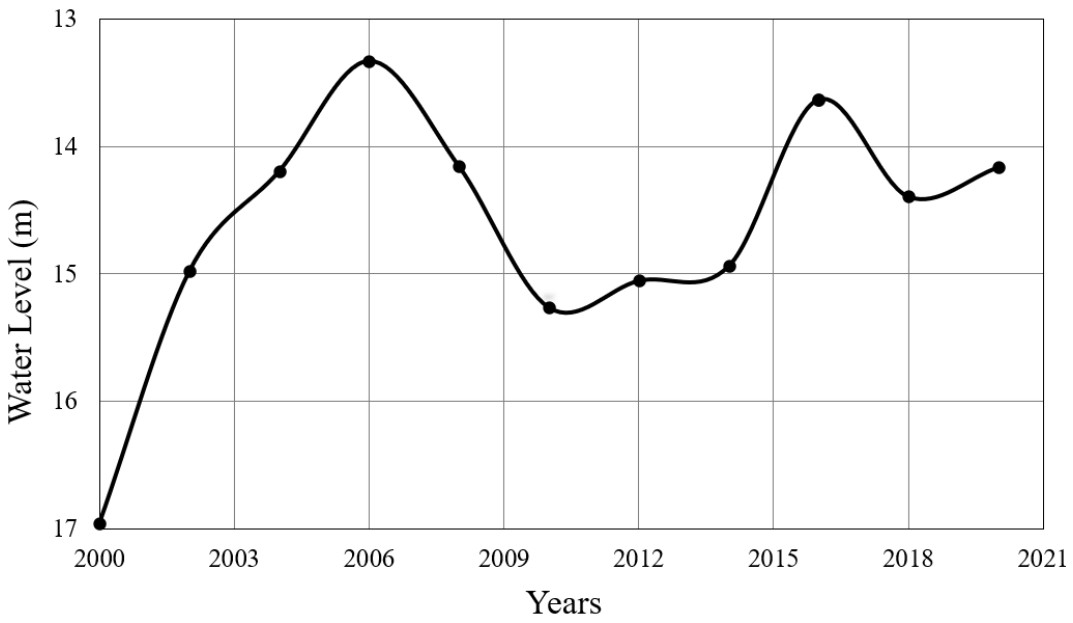

**Figure 6.** Time-series graph showing the average water level variation in Victoria County over the past 21 years.

### 3.3. Optimized Hotspot Analysis

The optimized hotspot analysis revealed three clusters in the study area: Cluster-1, Cluster-2, and Cluster-3 (Figure 7). Cluster-1 shows the high concentration of oil wells in the western part of the County. Cluster-2 and Cluster-3 delineate the high concentration of wells in the southern part of the County. The annual oil and gas production data were plotted to produce a time-series graph (Figure 8) that shows high extraction of oil (~1,900,000 BBL) and gas (~1,300,000 MCF) from 2017 to 2021 marked by the red rectangle in Figure 8.

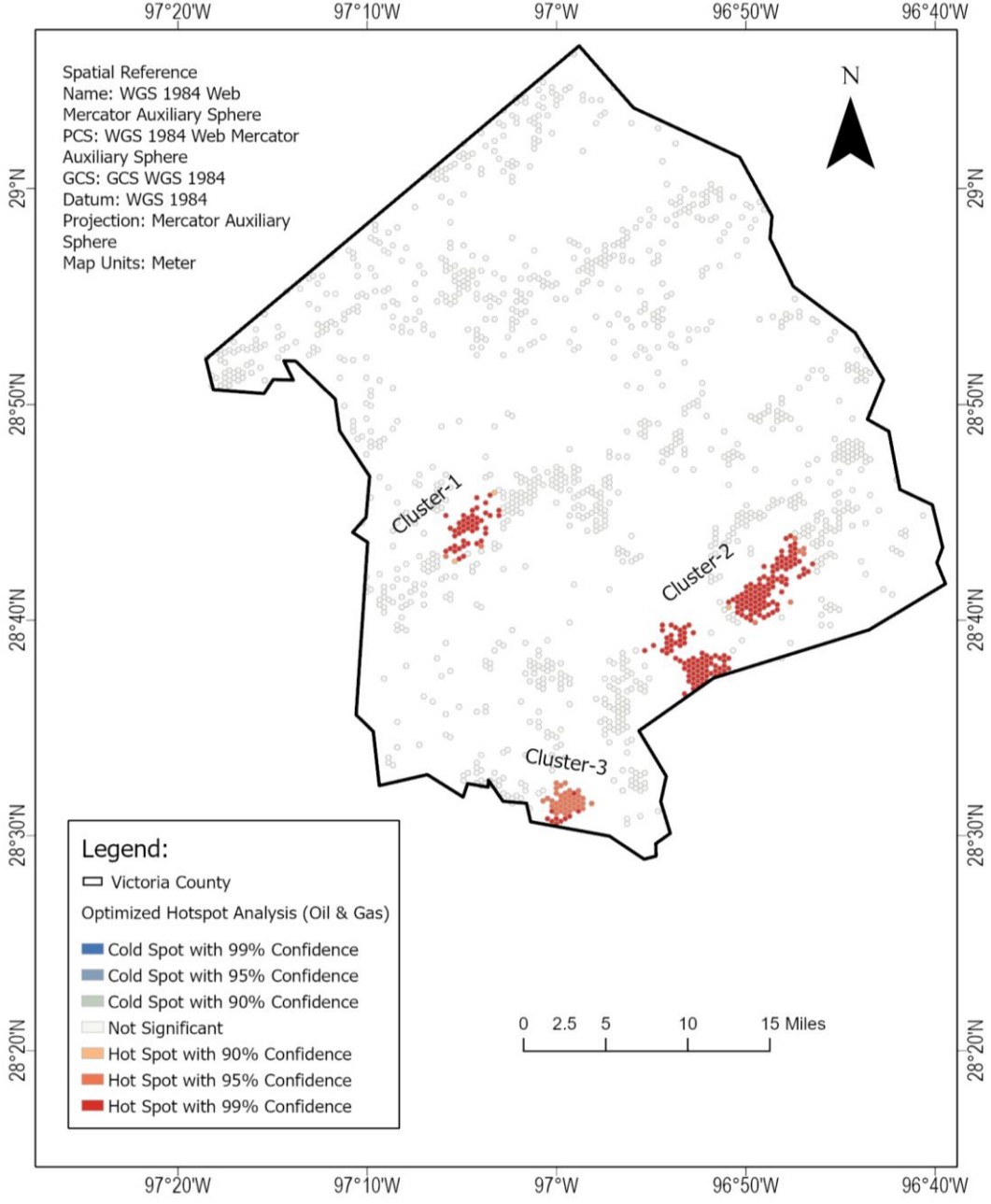

**Figure 7.** Optimized hotspot analysis, depicting the high concentration (cluster-1, cluster-2, and cluster-3) of oil and gas wells over the County.

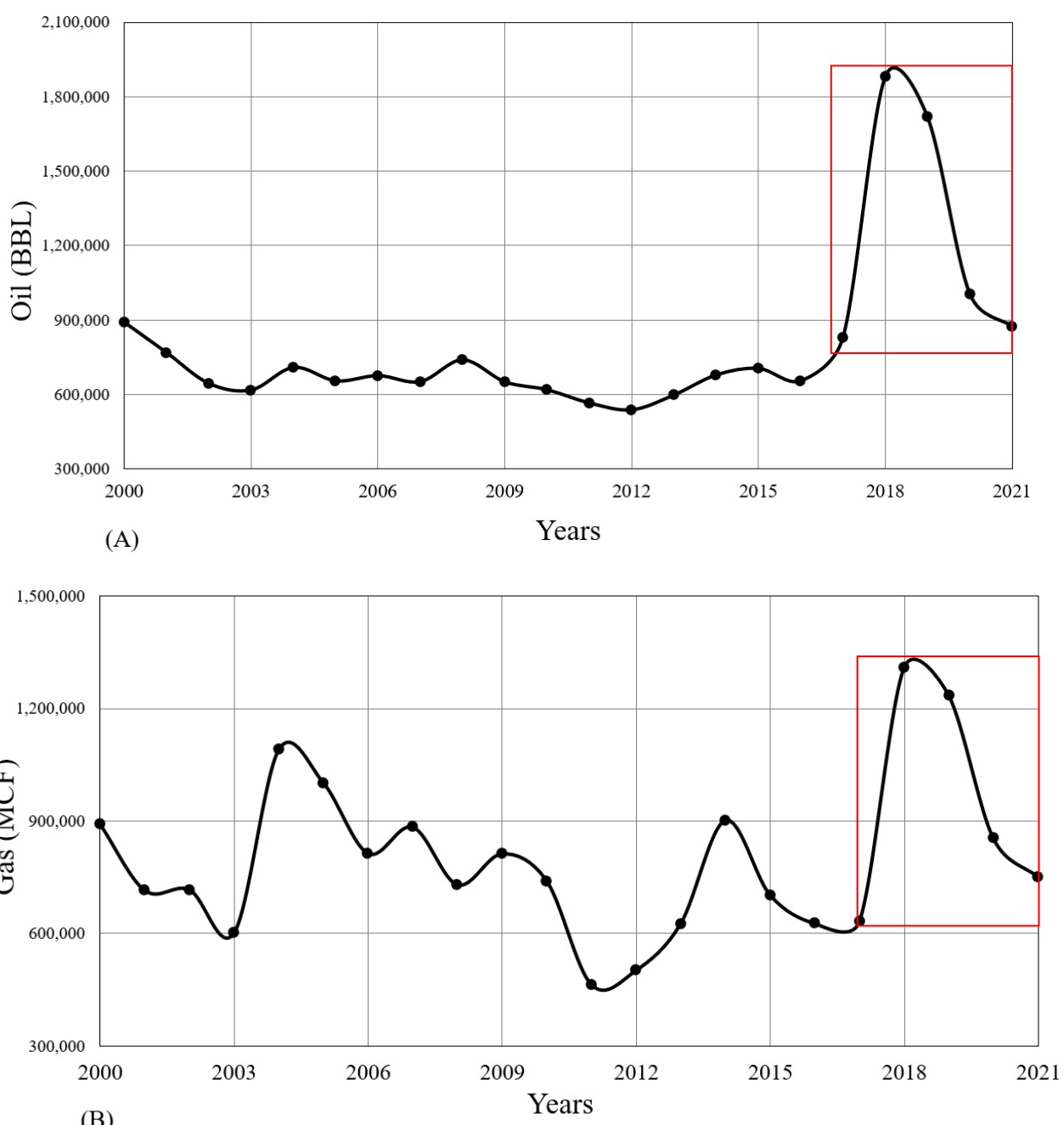

**Figure 8.** (**A**). Annual time-series graph of annual oil production in Victoria County over 21 years. The red rectangle depicts high extraction of oil from 2017 to 2021. (**B**). Annual time-series graph of gas production data in Victoria County over 21 years. The red rectangle shows the high extraction of gas from 2017 to 2021.

## 4. Conceptual Models

Conceptual Model-1 is our optimum consideration of the natural groundwater flow system in the study area (Figure 9). It represents how recharge, discharge, groundwater, surface water communications, and cross-formational flow take place inside the aquifers and within the confining units of a flow system. The impact of the overexploitation of fluid resources such as groundwater, gas, and oil is shown in the cross-section of the conceptual model in the northwest–southeast direction. The agricultural, industrial, and domestic groundwater usage increased rapidly during the 20th century in Gulf Coast counties. Groundwater is withdrawn from the three aquifers in semi-confined and confined portions

of the Gulf Coast aquifer system and oil is withdrawn from deep boreholes (northern part) and offshore (Gulf of Mexico). The hydraulic pressure on the sediments decreases, causing the de-watered sediments to compact due to the weight of the overlying sediments. The clays compact due to the reduced internal pressure in the clays and the overburden, resulting in land-surface subsidence. If pumping rates are low, this will have little effect because sand and clay layers are de-watered first and these compact only slightly. The variability of climatic parameters, climate change, and the tectonic effect (normal faults) may have important roles in affecting land subsidence in the Victoria area.

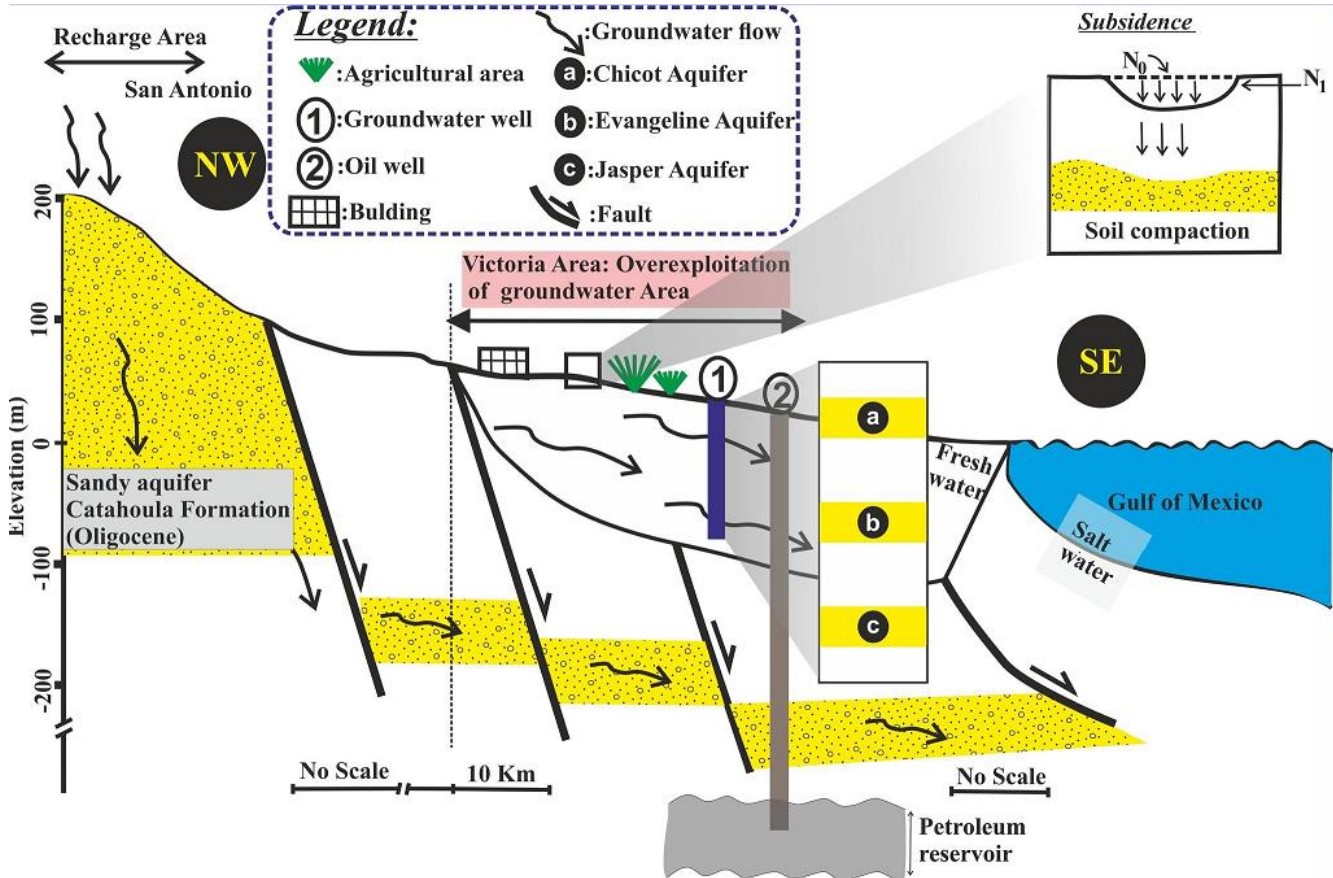

**Figure 9.** Conceptual Model-1 shows the connection between the surface water and groundwater, and the impact of overexploitation.

Conceptual Model-2 demonstrates that if fluid extraction continues, groundwater will start to be drawn from less transmissive clay levels (Figure 10). While sand grains are round, clay particles are sheet-like. As they become de-watered and compacted, they align perpendicular to the load applied by overlying sediments. As clay particles line up in the same direction, the permeability, porosity, and thickness of the clay layer decrease. The swelling and drying of clays in the study area can disturb its fluid hydrodynamics and cause serious problems such as piezometric anomalies. The reservoir can lose its reservoir characteristics due the sediment compaction, which may present an increasing challenge over time in the study area.

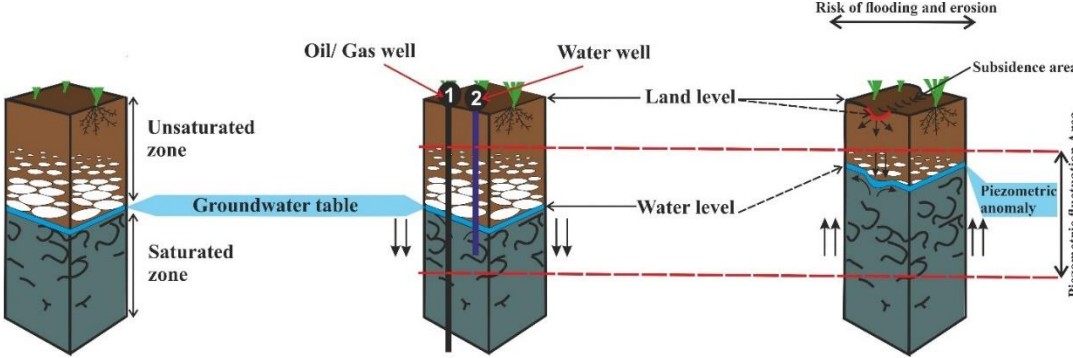

**Figure 10.** Conceptual Model-2 reveals that the overexploitation of oil/gas is causing sediment compaction that leads to land subsidence and risk of flooding/erosion.

## 5. Discussion

### 5.1. Groundwater Change

Land subsidence due to water level decline has been occurring in many parts of the world. One such region is along the Gulf Coast of Texas, covering Victoria County and the Houston–Galveston area [1,62]. In Victoria County, groundwater is the key water source for households, agriculture, and industry. In potentiometric surface maps, the variation of the groundwater level in the County was initially observed to be low, but this decline in groundwater level changed. A time-series graph is beneficial for determining a change in the groundwater level. Groundwater level decline that was observed over the period of 2006–2015, both in the potentiometric maps and the time-series graph, could have affected the stability of the area. A drought period was reported earlier during 2005–2006 and 2007–2009, and the most prolonged drought was observed during 2010–2015 along the Gulf Coast [63]. During the 2005–2006 drought event, the aquifer storage was as low as $-14$ km$^3$, while during the 2007–2009 drought event, the lowest storage was assessed to be $-7.4$ km$^3$ [63]. The most important groundwater decline happened during the 2010–2015 drought period, with a water volume change of $-3.38 \pm 0.43$ km$^3$·yr$^{-1}$. The aquifer storage reached a very low level in this time period due to the low precipitation and high pumping rate [63].

Withdrawal of groundwater is one of the reasons for land subsidence due to the resulting compaction of aquifer systems [64]. The two types of ground motion that typically happen in such susceptible aquifer systems are (1) deformation and (2) ground failures. Deformation caused by vertical and horizontal movement of the land surface is the leading risk related to fluid extraction. Ground failures, such as earth fissures and reactivation of surface/subsurface faults (growth faults), can be associated with area vertical ground displacement [65]. Therefore, a drainage model with MODFLOW simulation competencies will be valuable for regional simulations of groundwater flow, aquifer system compaction, and land subsidence in this study area.

The emerging hotspot analysis performed in this study showed that the groundwater level transition is in the northern part of the County and around Victoria City. The overall trend of the groundwater level in the County was an increase, but the major groundwater decline (2006–2015) may be a contributing factor to this area's subsidence. In further studies, 3D finite element numerical modeling can be applied to predict the groundwater level and land subsidence in future pumping situations [66].

### 5.2. Hydrocarbon Extraction

Hydrocarbon withdrawal is another contributing factor to land subsidence. Major subsidence (110 mm/year) in the Goose Creek Oil Field, Houston, has been reported as a result of oil/gas pumping [67]. The high hydrocarbon production in this area may induce fault reactivation [68,69] and reservoir compaction [70]. Such reservoir compaction due to

production can substantially influence the surface/subsurface subsidence [71]. Detailed studies and modeling, integrated with other mechanisms, are needed to precisely assess the interaction between these activities and their combined contributions to subsidence in this area. In this study, the results of the optimized hotspot analysis performed on the hydrocarbon wells of Victoria County showed a higher number of oil/gas wells in the central and southern parts of the study area than elsewhere. In the annual time-series graph, over the period of 2000–2021, high hydrocarbon extraction was noted during 2017–2021, which could contribute to the land subsidence in this region. The high hydrocarbon production reduces the pressure of reservoirs in the study area, and pressure changes affect the original stress field through poroelastic coupling [72,73]. The rate of compaction at the reservoir level and subsidence are mutually dependent. Forward modeling can be used if the amount of the reservoir compaction is known, or this can be predicted within an acceptable confidence level, along with when the existing or future subsidence must be assessed. Numerous authors have investigated subsidence triggered by hydrocarbon extraction and proposed approaches for subsidence prediction [74].

## 6. Conclusions and Recommendations

In this study, we used geospatial analysis and conceptual models to evaluate and correlate the groundwater level and oil and gas extraction with the land subsidence in Victoria County. The groundwater level has become shallower in the northwest region (32–22 m) and in the southeast region (14–6 m), with notable decline, as perceived from the potentiometric surface maps. The annual time-series graph further corroborated the results of potentiometric surface maps. It showed that the overall groundwater level has become shallower in the past 21 years, with a notable decline period (2006–2016) due to a drought that may have contributed to subsidence in the study area. The emerging hotspot analysis showed new, intensifying, sporadic, and persistent hotspots in the northwest region, and persistent and intensifying coldspots in the southeast region. The optimized hotspot analysis then revealed a high concentration of oil and gas wells in the southern region of the County and a high level of extraction of oil (1,900,000 BBL in 2018) and gas (1,300,000 MCF). The conceptual models correlated the water, hydrocarbon extraction, and sediment compaction with subsidence and suggested that the study area's aquifers may permanently lose their characteristics.

To avoid this phenomenon, we suggest:

- controlling the overexploitation of water and pumping of oil and gas;
- minimizing hydrocarbon exploitation or use injection to avoid more subsidence of land and saline intrusion of aquifers;
- conducting a study of the vulnerabilities of coastal aquifers;
- better planning for the management, development, and sustainability of these coastal aquifers;
- simulation modeling of these aquifers using MODFLOW and computational methods.

**Author Contributions:** Conceptualization, M.Y. and Y.H.; methodology, M.Y.; software, M.Y.; validation, M.Y. and M.Q.; formal analysis, M.Y.; investigation, M.Y.; resources, S.D.K.; data curation, S.D.K.; writing—original draft preparation, M.Y.; writing—review and editing, Y.H. and M.Q.; visualization, M.Q.; supervision, Y.H.; project administration, M.Y.; funding acquisition, S.D.K. All authors have read and agreed to the published version of the manuscript.

**Funding:** No specific funding was received for this work. Software and facility of GEORS lab were used for this work. Fulbright Fellowship supported YH visit to UH.

**Data Availability Statement:** I have included all data in my main manuscript file.

**Acknowledgments:** Higher Education Commission of Pakistan Fellowship is supporting MY's studies at the University of Houston. MY benefited with discussion with Ozzy Tirmizi and initially help form Otto Gadea. The authors express their grateful thanks to the reviewers who generously contributed their time and expertise to improve the quality of the manuscript.

**Conflicts of Interest:** The present paper is an original work, and all the authors declare that they have no conflicts of interest. The authors confirm that they are not associated with or involved in any profitable organization or company that has any financial interest.

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
