# Peer review of "Assessing Impacts of Land Subsidence in Victoria County, Texas, Using Geospatial Analysis"

_land, doi:10.3390/land11122211_

Round 1
Reviewer 1 Report
First, the title includes " Management, development, and sustainability of the Gulf Coast aquifer ", but it is not highlighted in the Results and Discussion section. Second, in “1. Introduction”, it should be divided into several paragraphs according to different contents, not all contents are one paragraph. Multiple paragraphs make it easier for readers to understand and read. And the specific description of the study area should be better placed in "Materials". Thus, the content of "2. Materials and Methods" needs to be reorganized to clarify the differences between the materials (e.g. study area and data) and methods (e.g. the methodology framework and formulas). I suggest that the authors focus on revising "1. Introduction" and "2. Materials and Methods".
Some specific changes are as follows:
1. Line 51 “The coastal part of Texas, besides the Houston Metroplex, is also suffering from land…”
The description of "The coastal part of Texas" is suggested to be a separate paragraph for better understanding by the reader. And add the status of research on the Gulf Coast aquifer.
2. Line 56 “Geospatial analysis is a valuable technique that allows for”
The advantages of geospatial analysis are illustrated in the introduction by adding the common methods currently used for ground subsidence studies.
3. Lines 78-102 “Victoria County is……in various strata of the Gulf Coast aquifer system [40].”
This content should be presented as a new paragraph. The introduction should state the reasons for choosing Victoria County, such as the severity or typicality of land subsidence in the region. Details of the study area would be more appropriately described in "2. Materials and Methods", with "Study Area" as a subheading.
4. Lines 101-102 “This variation, along with growth faults in this area, factored into the heterogeneity, currently seen in various strata of the Gulf Coast aquifer system [40].”
In the last paragraph of the introduction, please describe the innovation of this research, or the significance of the research.
5. Line 103 “2. Materials and Methods”
This title should contain 2 parts, “Materials” should contain a description of the study area, the data used, and “Methods” contains the study idea, the technical path and the specific methodology.
6. Line 111 “Figure 1.”
Add the extent of Gulf Coast aquifer to Figure 1.
7. Line 122
All formulas look vague, please check.
8. Lines 160-161 “To observe the rates of subsidence/uplift in the study area, the results of five GNSS stations covering 20 years are obtained from the Nevada Geodetic Laboratory.”
It is recommended to count the source and specific information of all data in Methods for better understanding by the reader, e.g. a sheet.
9. Line 158 “Figure 3”
Figure 3 was the method framework, which should be placed at the beginning of the Methods section.
10. Lines 166-167 “Based on the data available and the results obtained, two groundwater conceptual models, Model-1 and Model-2…”
Please explain the details of Model 1 and Model 2, the design steps, etc.
11. Line 220 “Figure 3. (a-e)…(u-v)”
Please check the serial number of all figures, which should be Figure 4. It is recommended that the multiple small figures in "3.2 Emerging Hotspot Analysis", currently "Figure 3", be combined into one figure.
12. Line 236 “…graph (Figure 5) of average water level was estimated for the whole Victoria County…”
Figure 5 not found.
13. Line 252 “…in the southern part of the county (Figure 7).”
The figure referenced in the current section should be placed after the most recent paragraph of text; please reorder the figures.
14. Lines 298-301 “Figure 8.- Figure 9.”
It is recommended to merge the 2 figures.
Author Response
- Line 51 “The coastal part of Texas, besides the Houston Metroplex, is also suffering from land…”
The description of "The coastal part of Texas" is suggested to be a separate paragraph for better understanding by the reader. And add the status of research on the Gulf Coast aquifer.
Answer: Thank you for this suggestion. We have moved the text to the next paragraph.
- Line 56 “Geospatial analysis is a valuable technique that allows for”
The advantages of geospatial analysis are illustrated in the introduction by adding the common methods currently used for ground subsidence studies.
Answer: We are unsure if we are asked to make any changes. But we added this discussion to highlight the importance of geospatial analysis. Specifically, this study utilized optimized and emerging hot spot analyses
- Lines 78-102 “Victoria County is……in various strata of the Gulf Coast aquifer system [40].”
This content should be presented as a new paragraph. The introduction should state the reasons for choosing Victoria County, such as the severity or typicality of land subsidence in the region. Details of the study area would be more appropriately described in "2. Materials and Methods", with "Study Area" as a subheading.
Answer: The reason for choosing Victoria County as our study area has been mentioned in lines 64 – 70. Also, the details of the study area have been moved under Materials and Methods lines 78-98. Thank you for this valuable input.
- Lines 101-102 “This variation, along with growth faults in this area, factored into the heterogeneity, currently seen in various strata of the Gulf Coast aquifer system [40].”
In the last paragraph of the introduction, please describe the innovation of this research or the significance of the research.
Answer: This is a fair point to be raised. Thank you. We have added significance of the research (Lines 71-76)
- Line 103 “2. Materials and Methods”
This title should contain 2 parts, “Materials” should contain a description of the study area, and the data used, and “Methods” contains the study idea, the technical path, and the specific methodology.
Answer: Study Area has been brought under the Materials subheading. Methods subheading has also been introduced. We appreciate the suggestion for improvement in this section.
- Line 111 “Figure 1.”
Add the extent of the Gulf Coast aquifer to Figure 1.
Answer: Thank you for this valuable input. The extent of the Gulf Coast aquifer in Figure 1 has been added.
- Line 122
All formulas look vague, please check.
Answer: Thank you for this comment. We went through all the formulas and found them correct.
- Lines 160-161 “To observe the rates of subsidence/uplift in the study area, the results of five GNSS stations covering 20 years are obtained from the Nevada Geodetic Laboratory.”
It is recommended to count the source and specific information of all data in Methods for better understanding by the reader, e.g. a sheet.
Answer: This is a fair point to be raised. We removed this data and are using recently published InSAR data for this area (Lines 64-70). Thank you for the suggestion.
- Line 158 “Figure 3”
Figure 3 was the method framework, which should be placed at the beginning of the Methods section.
Answer: Figure 3 has been placed at the beginning of the Method section as per your suggestion. We appreciate the improvement in this section.
- Lines 166-167 “Based on the data available and the results obtained, two groundwater conceptual models, Model-1 and Model-2…”
Please explain the details of Model 1 and Model 2, the design steps, etc.
Answer: Thank you for the nice suggestion. The design steps for Model 1 and Model 2 included collection of aquifer data (lithology, thickness, the direction of water flow, etc.), tectonics, type of drilling (water or oil)
- Line 220 “Figure 3. (a-e)…(u-v)”
Please check the serial number of all figures, which should be Figure 4. It is recommended that the multiple small figures in "3.2 Emerging Hotspot Analysis", currently "Figure 3", be combined into one figure.
Answer: We apologize for the inconvenience caused by the incorrect number of figures in the manuscript. We have fixed the names and numbers of the figures as well as their position in the text. We have double-checked the details of the figures. We highly appreciate your thorough review of the manuscript.
- Line 236 “…graph (Figure 5) of average water level was estimated for the whole Victoria County…”
Figure 5 not found.
Answer: We apologize for the inconvenience caused by the incorrect number of figures in the manuscript. We have fixed the names and numbers of the figures as well as their position in the text. We have double-checked the figures' details to avoid any confusion. We highly appreciate your thorough review of the manuscript.
- Line 252 “…in the southern part of the county (Figure 7).”
The figure referenced in the current section should be placed after the most recent paragraph of text; please reorder the figures.
Answer: We apologize for the inconvenience caused by the incorrect number of figures in the manuscript. We have fixed the names and numbers of the figures as well as their position in the text. In addition, we have double-checked the details of the figures. We highly appreciate your thorough review of the manuscript.
- Lines 298-301 “Figure 8.- Figure 9.”
It is recommended to merge the 2 figures.
Answer: Thank you for this suggestion. We have merged these two graphs.

Reviewer 2 Report
very interesting contribution to the discussion of environemental impacts of racking and exploiting of groundwater resources!
Author Response
A very interesting contribution to the discussion of the environmental impacts of racking and exploiting of groundwater resources!
Answer:
Thank you very much for your appreciation. We have revised the manuscript and improved it further

Reviewer 3 Report
The manuscript is interesting.
The subject of the manuscript is consistent with the scope of the Journal.
The abstract faithfully conveys the scope of investigations and conclusions drawn. The keywords correspond to the scope of the research.
I think the paper needs some corrections:
1) the abstract of paper is too long,
2) reduce keywords according to Instructions for Authors,
3) format the manuscript according to Instructions for Authors.
Author Response
The manuscript is interesting. The subject of the manuscript is consistent with the scope of the Journal. The abstract faithfully conveys the scope of investigations and conclusions drawn. The keywords correspond to the scope of the research.
- I think the paper needs some corrections:
- the abstract of the paper is too long
Answer: Thank you for pointing this out. The abstract has been considerably shortened
- reduce keywords according to Instructions for Authors,
Answer: We kept the keywords within the range given in the “Guidelines for Authors” of the journal.
- format the manuscript according to the Instructions for the Authors.
Answer: Thank you for highlighting this point. We have tried our best to update the manuscript according to the journal format.

Reviewer 4 Report
The paper submitted is well organized, and the subject of interest, with a nice application of the hotspot analysis.
Still, it needs some minor language revision (some examples are lines 90-92, 353-362).
Do the authors have established any clear effects of land subsidence on the population?
For future approaches, do the authors consider a validation of hotspots or coldspots using LIDAR technology?
Author Response
The paper submitted is well organized, and the subject of interest, with a nice application of the hotspot analysis.
Still, it needs some minor language revision (some examples are lines 90-92, 353-362).
Do the authors have established any clear effects of land subsidence on the population?
For future approaches, do the authors consider a validation of hotspots or coldspots using LIDAR technology?
Answer: Thank you very much for your appreciation. We thoroughly revised this paper and we haven’t developed any population effects on the subsidence since the focus of this paper has been the causes of land subsidence using the data integration technique.

Round 2
Reviewer 1 Report
Some minor modifications are suggested as follows.
1) It is recommended that the figures be placed after the paragraph where it is cited for better understanding. For example, Figure 1 can be placed after Line86.
2) A description of the research framework should be added in the first paragraph of “2.2 Methods”.
